# Addressing Patient Requests to Add Dietary Supplements to Their Cancer Care—A Suggested Approach

**DOI:** 10.3390/nu15245029

**Published:** 2023-12-07

**Authors:** Moshe Frenkel, Meroe B. Morse, Santhosshi Narayanan

**Affiliations:** 1Complementary and Integrative Medicine Service, Oncology Division, Rambam Health Care Campus, Haifa 3109601, Israel; 2Department of Family Medicine, The University of Texas Medical Branch, Galveston, TX 77555, USA; 3Section of Integrative Medicine, Department of Palliative, Rehabilitation, and Integrative Medicine, The University of Texas MD Anderson Cancer Center, Houston, TX 77030, USA; mbmorse@mdanderson.org (M.B.M.); snarayanan2@mdanderson.org (S.N.)

**Keywords:** complementary medicine, alternative medicine, integrative medicine, cancer care, oncology, nutritional supplements, natural products, dietary supplements, unmet needs, patient-centered care, patient–doctor communication, compassion, empathy

## Abstract

Dietary supplements are widely utilized by cancer patients as part of a complementary and integrative approach to their healthcare. However, a significant portion of patients refrain from discussing their supplement use with their physicians, often due to the perceived indifference or negativity of their healthcare providers. This communication gap exposes patients to unreliable information sources and potential risks associated with uninformed supplementation. As the healthcare landscape evolves, there is an increasing recognition of the pivotal role that physicians play in guiding patients’ healthcare decisions. A patient-centered perspective prioritizes the provision of evidence-based information tailored to the individual’s needs. It advocates for open discussions about potential risks and fosters shared decision making, respecting patient autonomy. Additionally, this approach involves offering alternative options, documenting patient preferences, and ensuring ongoing support while coordinating with the healthcare team. To address these evolving needs, healthcare providers must adopt a transformative perspective, becoming expert guides who engage with their patients as informed and empowered participants. This revised approach emphasizes an open dialogue that balances presenting facts and acknowledging uncertainties surrounding dietary supplement use. Our narrative review of the literature underscores the importance of a practical approach, centered on transparent discussions and respect for patient autonomy. By following this approach, healthcare providers can empower patients to navigate the complexities of dietary supplement use within the context of cancer care, thereby safeguarding patient safety and overall well-being. Notably, our proposed tool highlights the utilization of reliable sources, the risk stratification of supplements, specific recommendations, and subsequent monitoring, providing a structured framework for informed decision making.

## 1. Introduction

A recent consumer survey conducted by the Council for Responsible Nutrition (CRN) unveiled an interesting trend in American healthcare habits: approximately three-quarters of Americans incorporate dietary supplements into their daily routines, with many users attributing the use of these supplements to their overall well-being [1].

Dietary supplements (DSs) represent a category of products designed to augment one’s diet by providing dietary elements, such as vitamins, minerals, herbs or botanicals, amino acids, and other such constituents. These supplements come in various forms, such as pills, capsules, tablets, and liquids [2]. 

Among the diverse array of complementary and integrative therapies adopted by cancer patients, DSs rank among the most accessible and widely embraced. Previous studies have suggested that anywhere from 20% to 90% of individuals affected by cancer turn to these products as part of their healthcare regimen [3,4].

Despite their common use, patients tend to shy away from discussing their DS usage with their healthcare providers [5,6].

Moreover, healthcare professionals have voiced concerns about unsupervised supplement use potentially leading to adverse effects and unfavorable interactions, particularly in the context of cancer patients undergoing active treatment. For instance, according to animal studies in the Physician Data Query [7], vitamin C has been found to reduce the bioavailability of imatinib. Additionally, animal studies have indicated that vitamin C might hinder apoptosis when used alongside medications like doxorubicin, cisplatin, vincristine, methotrexate, and imatinib [8]. Pre-clinical research has also shown that EGCG, or green tea extract, can potentially counteract the anti-cancer effects of bortezomib. It is essential to note that most of these adverse interactions have only been observed in animal studies [7]. 

Regrettably, our current understanding of the effectiveness of DSs in cancer care remains limited, with only a handful of benefits substantiated through clinical trials. A systematic review from 2012 suggested that there were insufficient data to endorse the use of DSs in westernized populations, although vitamin D and omega-3 fatty acids were notable exceptions [9]. In a more recent development, the 2018 World Cancer Research Fund/American Institute for Cancer Research Report cast doubt on the potential of DSs to enhance the prognosis or overall survival after a cancer diagnosis, drawing on evidence from observational studies and clinical trials [10]. According to this report, the primary course of action should be to obtain essential nutrients from dietary sources, with supplements to be considered only when a deficiency is demonstrated either biochemically (e.g., low plasma vitamin D levels, B12 deficiency) or clinically (e.g., low bone density). The authors added that DSs should also be contemplated when nutrient intake persistently falls below two-thirds of the recommended levels. The report unequivocally discouraged the use of supplements for cancer prevention.

Given this consensus among experts about the lack of evidence supporting the use of DSs for cancer patients, and the potential risks associated with their use alongside cancer treatments, the question arises as to why DSs remain popular among patients.

In one study, nearly 40% of patients attending an integrative oncology clinic in a major comprehensive cancer center said they arrived with the intention of discussing their supplement usage [11]. Of these patients, 75% were already using vitamins, and nearly 30% were using herbs before their consultation, all with the expectation of addressing these matters with their physicians.

In a more recent study at another comprehensive cancer center, approximately 50% of adult patients with breast, colorectal, lung, or prostate cancers undergoing active treatment reported using various DSs [12]. 

Furthermore, numerous studies have indicated that even cancer survivors are more inclined to use DSs [13,14]. In one study, the prevalence of DS use was notably higher among American cancer survivors (70.4%) compared to cancer-free individuals (51.2%) [13]. 

Recent research has offered the intriguing possibility that these DSs might confer some beneficial effects to cancer patients [15,16]. These findings have suggested that modifying certain nutrients in one’s diet could influence the effectiveness of cancer therapies and alleviate cancer-related symptoms [15]. At times, DSs may even enhance the efficacy of cancer treatments [16]. 

Nutrients can influence epigenetic states through mechanisms that include DNA methylation, histone modifications, and miRNA-dependent gene silencing [17]. These alterations have been associated with either an increased or a decreased risk of cancer development. There is compelling evidence that certain foods play a protective role in cancer prevention by directly inhibiting tumor progression, or by altering the tumor’s microenvironment to create unfavorable conditions for tumor initiation or growth [16,17].

This body of evidence indicates that patients may have valid reasons for exploring DS usage. Consequently, there is a growing need to provide well-informed guidance on this issue in an oncology setting.

To unlock the potential benefits of DSs, additional clinical studies are required to determine the optimal dosages, most effective administration methods, and the bioavailability of various supplements. This ongoing research could validate and maximize the beneficial effects that have already been observed.

In this paper, we shall explore patients’ motivations for using DSs and how their use might affect patient–doctor communication, and we will describe a possible approach for addressing the need for guidance on this issue, which has been expressed by patients and families affected by cancer.

## 2. Motivations for Supplement Use

Understanding the motivations that drive patients to adopt dietary supplements (DSs) is pivotal, especially in the context of cancer care. These motivations provide valuable insights into why patients choose to integrate DSs into their journey with cancer.

One of the primary motivations for using DSs among cancer patients is the management of treatment side effects [5,6,18,19,20,21,22,23,24]. These side effects encompass issues like nausea, fatigue, and digestive challenges often associated with treatments such as chemotherapy, radiation, and immunotherapy. Additionally, patients dealing with anxiety tend to rely more on DSs, seeking psychological support and a sense of empowerment to take charge of their healthcare decisions [20]. For many, the objective of DS use is to enhance their quality of life rather than aiming solely for a cure [5,6,18,19,20].

Furthermore, cancer patients frequently turn to DSs with the aim of boosting their immune systems [5,6,18,19,20]. A robust immune response is crucial, helping patients fend off infections, recover from treatments, and maintain their overall health. Many view DSs, particularly those rich in vitamins, minerals, and antioxidants, as allies in fortifying their immune defenses.

Moreover, the fear of cancer recurrence is a prevalent concern among survivors [5,6,19,20]. To allay this anxiety, some individuals resort to DSs, believing in their anti-cancer properties, which they perceive as a protective shield against disease recurrence, and which offer a sense of security and control.

The pursuit of control and empowerment stands as a significant driver of DS usage [5,6,19,20]. A cancer diagnosis can often leave patients feeling adrift in a sea of uncertainty. DSs serve as a means of active participation in healthcare decisions, providing patients with a sense of control and autonomy over their health.

Improving the overall quality of life remains a central objective in using supplements [5,6,19,20]. Patients seek enhancements not only of their physical well-being, but also of their emotional and mental health. The holistic approach offered by supplement use, addressing the body, mind, and spirit, presents an attractive option for enhancing overall well-being.

Cultural factors and deeply ingrained beliefs significantly influence DS use [22,23,24]. Cultural practices and traditions often encourage specific supplement usage for health and healing, which patients tend to respect and follow.

Additionally, the allure of a natural and holistic approach to health often steers patients toward DSs. Faced with a life-altering diagnosis, patients view supplement use as an act of self-care, a form of self-advocacy that allows them to take charge of their health and play an active role in their treatment journey.

Patients source their information from various channels, including friends, family, the internet, and popular media, which influences their decisions [5,6,19,20]. The credibility and accuracy of these sources vary widely, contributing complexity to the decision-making process.

Recognizing and comprehending these motivations is essential for healthcare providers. It enables them to approach DS discussions with empathy, to engage in productive conversations, address patient concerns, and guide patients toward safe and informed choices. Effective communication should take into account the multifaceted motivations behind supplement use, and aim to support the holistic well-being of patients [19].

## 3. Challenges in Physician–Patient Communication

Effective communication between physicians and patients with cancer regarding the use of DSs is a multifaceted process fraught with several challenges. These challenges can significantly affect the patient–provider relationship, the quality of care, and patient outcomes [19].

One fundamental challenge is the limited knowledge many physicians have about DSs. Medical education primarily focuses on conventional treatments and pharmaceuticals, leaving healthcare providers ill-equipped to discuss DSs with their patients. This knowledge gap results in a lack of guidance on this topic. A survey of US oncologists revealed that not even half discussed DS use with their patients [23]. 

On the patient side, there is often a reluctance to disclose supplement use to their physicians [21]. This reluctance can stem from a variety of factors, including a fear of judgment or disapproval, concerns about the physician’s lack of knowledge about supplements, a perception that conventional medicine providers do not want them to use alternative therapies, or sometimes even a fear that their physician will discontinue the conventional cancer therapies [18,19,20]. This lack of openness leads to a breakdown in the exchange of information, with physicians often unaware of their patients’ supplement usage [18].

The absence of robust scientific evidence about DS use poses a substantial challenge. Supplements often lack the same level of rigorous research and safety data required of pharmaceutical drugs. The lack of clinical trials and comprehensive evidence make it difficult for physicians to provide evidence-based recommendations or to make informed decisions about supplement use. Both patients and physicians may feel uncertain about the effectiveness and safety of specific supplements [5,6,18,19,20].

Moreover, clinicians and patients alike may have concerns about the out-of-pocket financial investment that DSs demand, potentially further straining patients’ resources and distracting patients from an evidence-based focus on a well-rounded, whole food, plant-based diet. Adult cancer survivors spend approximately USD 6.8 billion annually on DSs [24]. Further research on the safety and efficacy of DS use in patients with cancer is imperative to ensure a balanced cost/benefit ratio.

Time constraints in clinical settings can be a practical challenge. Physicians often have limited time during patient appointments. Discussing DSs can be time consuming, involving a detailed review of the patient’s supplement regimen, potential interactions with prescribed medications, and patient education. These time constraints may hinder comprehensive discussions, leaving patients feeling unheard and misunderstood.

In some healthcare systems, physicians may not receive reimbursement for discussing DSs or providing integrative care. This financial disincentive can lead to a lack of motivation to engage in in-depth conversations about supplements with patients, creating a disconnect between patient needs and healthcare provider priorities [19].

Differences in beliefs between patients and physicians can be a significant obstacle to effective communication. While patients may see supplements as a way to complement their conventional treatments and improve their well-being, physicians who do not share this perspective may struggle to effectively communicate or support these choices. These differing beliefs can result in misunderstandings and a lack of trust [19].

Patients often encounter information about DSs that is conflicting, confusing, or misleading. Physicians must help patients with the complex task of sifting through this information by clarifying misconceptions, and providing evidence-based guidance [5,6].

Physicians must also manage patient expectations. Patients may have high hopes for DSs, anticipating significant benefits. Balancing these expectations with realistic, evidence-based information about the potential benefits and limitations of supplements can be a delicate task [5,6,19].

Trust is built on the belief that physicians have their patients’ best interests at heart, and when patients feel their choices are not acknowledged or respected, trust begins to wane [19].

Patients may feel misunderstood by their healthcare providers. DS use is often deeply rooted in patients’ beliefs, experiences, and personal wellness philosophies. When physicians fail to appreciate these perspectives and the reasons behind supplement use, patients can feel alienated. This can lead patients to the belief that their healthcare providers lack the understanding needed to offer personalized care, further eroding trust.

A lack of open dialogue can create an atmosphere of secrecy and opacity within the physician–patient relationship. Patients might feel that their physicians are withholding information or are not providing a complete picture of their care. Such feelings of secrecy can breed mistrust, leading to doubts about the transparency and honesty of the healthcare provider.

A breakdown in trust can influence treatment decisions. Patients who perceive their physicians as disregarding their preferences or choices regarding DSs may become more hesitant to follow their treatment plans or make informed decisions about their care. This can have consequences for treatment adherence and outcomes and patients’ overall well-being [19,20].

Moreover, a breakdown in trust may drive some patients to seek alternative healthcare providers who are more receptive to discussing and supporting DS use. While this can provide patients with a sense of understanding and alignment with their values, it may not always lead to the best healthcare outcomes, especially if the alternative providers lack conventional medical training or fail to communicate with the patients’ physicians [5,6,19].

Balancing their patients’ need for autonomy with the need to make well-informed choices is a challenge for healthcare providers. Encouraging open discussions, providing evidence-based information, and engaging in shared decision making are vital steps for supporting patient autonomy. This balance ensures that patients may make autonomous, well-informed choices about DSs while prioritizing their safety and overall well-being [19].

## 4. Developing an Approach to Address Cancer Patients’ Requests to Incorporate Dietary Supplements

To respond effectively to the requests of cancer patients seeking to integrate DSs into their treatment, healthcare providers must embrace a patient-centered perspective that acknowledges the patient’s need for autonomy and empowerment. In today’s healthcare landscape, patients increasingly value physicians who recognize patients’ role as empowered participants in shaping their own healthcare choices. Physicians should be seen as informed intermediaries, expert guides, and consultants, with the patient at the center of decision making. The optimal approach involves discussing both the facts and uncertainties surrounding DS use with the patient, facilitating a mutually informed decision-making process [19].

In 2009, the Society of Integrative Oncology (SIO) issued guidelines for the inclusion of complementary medicine in cancer care, and these guidelines included recommendations concerning the use of nutritional supplements. The SIO advised that patients interested in using DSs should consult with trained professionals who could provide support, set realistic expectations, and weigh the potential risks and benefits [25].

Recognizing the importance of addressing patients’ requests to incorporate DSs into their care, the Clinical Practice Committee of the SIO, in 2012, identified the necessity of a comprehensive approach suitable for integrative oncology settings. This approach places the patient at the center, aiming to provide evidence-based information tailored to the individual patient’s needs, the discussion of potential risks, and to promote shared decision making. To make progress in this effort to effectively advise patients on the integration of DSs, the committee members compiled a list of 10 DSs that, at the time, were considered safe and had enough supporting evidence for their use to be discussed with patients. These DSs were deemed appropriate for integration in specific situations, and this list was relevant at the time of publication in 2012–2013 [5].

A prominent comprehensive cancer center in the United States recently introduced another innovative care model that involves the use of a dedicated pharmacist to offer guidance to patients on the use of DSs [26]. In this program, integrative medicine providers who are engaged in counseling patients to address unmet symptom needs prescribe DSs when deemed appropriate, considering the clinical context, patient preferences, and available research evidence, using the clinical pharmacist as an additional professional who helps with providing this advice [26]. To assess the feasibility and outcomes of this approach, researchers conducted a retrospective analysis using medical records. The results demonstrated the viability of integrating DSs into an academic oncology setting, showing high patient satisfaction levels and minimal adverse events [26].

Recognizing the evolving landscape of DS use in cancer care, we emphasize the need for a general approach to be provided by integrative oncologists. This approach should be adaptable for integration into oncology settings where additional professionals may not be available, and which rely primarily on the expertise of the integrative practitioner. Such an approach should be patient-centered, and encompass the provision of evidence-based information, tailored assessments of individual patient needs, discussions about potential risks, encouragement of shared decision making, respect for patient autonomy, the presentation of alternative options, the documentation of physician and patient preferences, and the provision of ongoing support, all while ensuring coordination with other members of the healthcare team. Staying informed and supporting informed and empowered choices are the key components of this comprehensive approach, which ultimately enables patients to make well-informed decisions that prioritize their safety and well-being in the context of cancer care.

When a patient requests the addition of DSs to their cancer care, healthcare providers must dedicate sufficient time to address the patient’s needs thoroughly. To provide an informed response, providers should use reliable sources, such as About Herbs from Memorial Sloan Kettering Cancer Center [27] or the Natural Medicine Database [28], to evaluate the safety and efficacy of the supplement in question, the potential side effects, drug–herb–nutrient interactions, and product quality. The findings from such searches can be divided into three categories based on the risk and benefit value (Figure 1):High risk and low or unknown benefit;Questionable risk and low or unknown benefit;Minimal risk and clinical clues regarding its benefit.

### 4.1. High Risk and Low or Unknown Benefit

If the research findings indicate that the use of a particular DS raises significant safety concerns, such as the potential for side effects and interactions, while the benefits remain unclear and are primarily based on speculation rather than clinical research, it is advisable to discourage such use. In such cases, healthcare providers should engage in a discussion that acknowledges the risks involved and also documents these concerns.

### 4.2. Questionable Risk and Low or Unknown Benefit

When the research findings reveal no known safety or interaction concerns based on clinical research, yet there is no concrete evidence of clinical benefits, patients should not be dissuaded from using a supplement. It is important to respect their choice, even if their motivations for using a supplement are not based on scientific evidence. In such situations, an open and honest discussion with patients about their motivations, as well as a detailed discussion of the advantages and disadvantages of supplement use, should take place. Given the absence of major safety concerns, the approach should be to empower patients to make their own informed decisions regarding the use of a supplement.

### 4.3. Minimal Risk with Clinical Clues Regarding the Benefit

When the research findings indicate that the risk associated with a supplement’s use is minimal, and there are clinical clues suggesting potential benefits, patients can receive support for this specific use. Even in this favorable scenario, healthcare providers should engage in an open discussion with their patients about the advantages and disadvantages of the supplement. This discussion should also include recommendations of high-quality products, if available, as well as a conversation about the optimal dosage. Additionally, a plan for further follow-up to monitor the use of the supplement should be established.

A discussion about DS use should take place with all patients who seek advice about DSs during and after their cancer care. Patient perspectives and expectations should be explored, and the inherent uncertainty in DS use should be discussed, including the pros and cons, for all three categories.

It is crucial to emphasize that even if patients choose to go against their advice, healthcare providers should maintain an open channel of communication, coupled with empathy and compassion. This approach will help to maintain communication and the monitoring of DS use, adverse reactions, and potential interactions, which is essential for intervention or treatment adjustments. Another important aspect of approaching DS use among cancer patients is informing the oncologist regarding the recommendations and patients’ decisions. This information will help the oncologist to understand the rationale behind supplement use and to help monitor for adverse effects or interactions, in case they need to add new medications to the treatment plan for the cancer. 

The physician’s role does not conclude with the initial discussion. Ongoing support and monitoring are vital as patients integrate DSs into their care plan. Collaboration with other healthcare team members, such as oncologists and nurses, is essential to promote patient safety and well-being, by ensuring that DSs do not interfere with the conventional treatments.

Remaining informed and updated is a commitment that physicians must uphold. The ever-evolving landscape of DSs and their role in cancer care demands a continuous dedication to learning. Ongoing education and staying informed about the emerging evidence are essential to providing patients with the most current and evidence-based information.

The ultimate goal is to empower patients to make well-informed and empowered choices. By encouraging and supporting patients in their quest to make informed decisions, physicians can ensure that patients are well equipped to navigate the intricate path of dietary supplement use in the context of cancer care. This comprehensive approach places patient safety and well-being at the forefront of healthcare decision making.

This suggested approach addresses cancer patients’ requests to integrate DSs into their care with a clear patient–doctor communication process relating to safety and efficacy, in a patient-centered manner.

## 5. Recommending Dietary Supplements as Food

Discussing nutrition with patients affected by cancer is one of the main issues discussed in the integrative oncology setting [29]. Nutrition and DSs, in many situations, are commonly discussed in the same encounter. Many DSs are derived from food, and although some DSs are combinations of multiple ingredients, not all are. Therefore, advising the patient to eat the whole food that the supplement is derived from is another strategy that might help meet the patient’s expectations. One example is the use of mushroom supplements for cancer. The data support an association between high dietary mushroom consumption and a lower risk of cancer [30]. Mushrooms are anti-inflammatory and immune-enhancing, and are often used in Asian countries during cancer treatment. However, due to the paucity of clinical data from many other countries, we suggest the dietary consumption of mushrooms that are accessible to the patients, instead of using them as supplements. Green tea is another example; its active ingredient, ECGC, has been found to be beneficial. However, ECGC consumption as a supplement has a risk of hepatotoxicity and, therefore, the dietary consumption of green tea is recommended. For supplements comprising several ingredients, the clinician should review the ingredients and recommend the use of their dietary form when possible.

## 6. Conclusions

In summary, effective communication between physicians and cancer patients regarding dietary supplement (DS) use stands as a cornerstone for establishing trust and ensuring optimal care provision. Acknowledging and actively addressing the inherent challenges to this communication process are pivotal for fortifying the patient–provider relationship, elevating care standards and, ultimately, leading to improved patient outcomes.

To meet these challenges and to achieve enhanced patient-centered care, healthcare providers must adopt a paradigm shift, viewing patients as informed participants in their healthcare decisions. Physicians should evolve into expert guides and consultants, valuing and respecting patient autonomy and empowerment. Our proposed approach advocates for engaging in an open dialogue that comprehensively considers both the factual information and uncertainties related to DSs, fostering mutually informed decision making.

Central to this recommended approach is the emphasis on open discussion, transparency, and the utmost regard for patient autonomy. By adhering to such principles, healthcare providers can effectively navigate the intricacies of discussing dietary supplement use in the realm of cancer care, prioritizing the safety and overall well-being of the patient. Furthermore, in implementing this approach, it is imperative to integrate reliable sources of information, engage in risk stratification, and offer specific recommendations tailored to the individual patient’s needs. Subsequent monitoring ensures continued support and guidance, further reinforcing patient safety and optimal health outcomes. This comprehensive approach not only addresses the challenges inherent in physician–patient communication regarding DSs, but also equips patients with the necessary knowledge and tools with which to make informed choices, thereby optimizing their cancer care journey.

## Figures and Tables

**Figure 1 nutrients-15-05029-f001:**
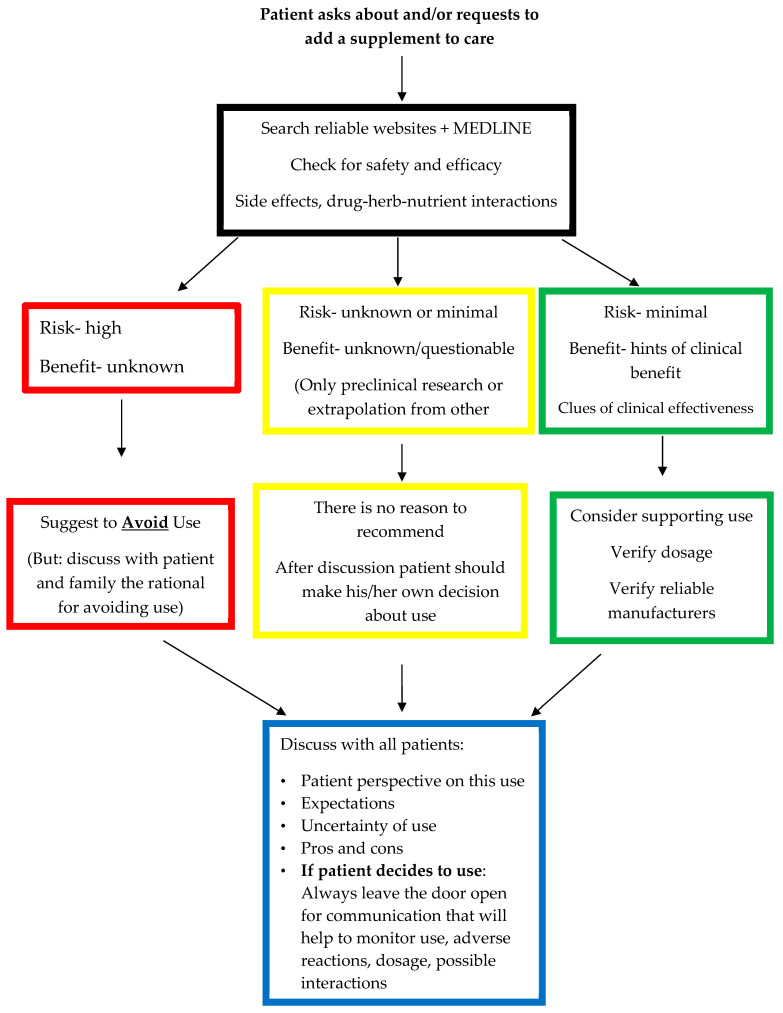
A suggested approach for addressing patients’ requests to add dietary supplements to their cancer care.

## Data Availability

Not applicable.

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
