# Peer review of "Addressing Patient Requests to Add Dietary Supplements to Their Cancer Care—A Suggested Approach"

_nutrients, 2023, doi:10.3390/nu15245029_

Round 1
Reviewer 1 Report
Comments and Suggestions for Authors
The objective of the review is clear—to emphasize a patient-centered perspective on discussing dietary supplement use in the context of cancer care.
This review promises to present a practical approach emphasizing open discussions, transparency, and respect for patient autonomy. Consider outlining the key components of this practical approach early in the manuscript for reader engagement.
Elaborate on specific skills or strategies that healthcare providers can adopt to become effective guides in the realm of dietary supplements. Discuss specific strategies or tools that patients can use to make informed decisions about supplement use.
The authors clearly identified various motivations for supplement use among cancer patients. Consider providing specific examples or anecdotes to illustrate each motivation, making the content more relatable. For each motivation, briefly discuss any scientific evidence supporting or questioning the effectiveness of supplements in addressing the specific motivation.
The review rightly emphasizes the role of supplements in managing treatment side effects. Discuss specific supplements or interventions that have shown efficacy in managing common treatment side effects like nausea, fatigue, or digestive issues
Provide insights into how healthcare providers can address the psychological aspects, offering support beyond the prescription of supplements.
The fear of recurrence as a motivation is discussed. However, please elaborate on the scientific basis or lack thereof regarding supplements' potential in preventing cancer recurrence.
The lack of physician knowledge about DS is a crucial challenge. Discuss potential solutions or strategies for addressing this knowledge gap, such as incorporating DS education into medical curricula or providing resources for physicians.
The suggestion for an optimal approach involving open dialogues and mutual decision-making is clear. Expand on how healthcare providers can integrate this approach into their practice, perhaps through training programs or guidelines?
Overall, the manuscript addresses an important aspect of patient care. Enhancing specific points and providing concrete examples or case studies could further strengthen the manuscript's impact and practical utility. While the motivations are well articulated, integrating scientific evidence and providing practical examples would enhance the comprehensiveness and applicability of the content.
Reviewer 2 Report
Comments and Suggestions for Authors
This is very interested and well-constructed paper, providing useful contextual information regarding the the dialogue between patients and clinicians regarding the use of dietary supplements in cancer care.
The papers takes a clear, logical approach through the different issues, and provides supporting information. I have a few minor suggestions that will hopefully improve the clarity of the paper:
1. It should be made clear (e.g. in line 24 of the abstract - "...we review the literature...") that the literature review is not systematic. I don't think it's a problem that it isn't systematic, it's just that it should be made clear.
2. When briefly discussing the potential effectiveness of DS in cancer care (e.g. line 56 onwards), it should also be noted that 'cancer' is not a single disease, and different tumours at different stages will respond very differently to DS.
3. In some cases in the background section, some statements seem to contradict others. For example, line 83 notes that "[DS was used by]... 70.4% of cancer survivors compared to 51.2% of cancer-free individuals", whereas the opening sentence of the whole paper says "...approximately three-quarters [i.e. 75%] of Americans incorporate DS into their daily routines". Likewise, line 57 states "A systematic review from 2012 suggested that there is insufficient data to endorse the use of DS in Westernized populations, although vitamin D and omega-3 fatty acids were notable exceptions", whereas line 84 states: "Recent research offers the intriguing possibility that these DS might confer some beneficial effects for cancer patients." I realise that - technically - this might not be a contradiction, but more information should be given to explain why.
4. In line 307, please could the authors provide more detail around what constitutes a "thorough" examination of the advantages and disadvantages of DS. What does "thorough" mean here?
5. Section 6 (line 349) is important, but I think this should be mentioned earlier. It feels almost like an afterthought here. The difference between whole food and supplements can be substantial, and it would be better to have this in the reader's mind before the whole of Section 5.
Reviewer 3 Report
Comments and Suggestions for Authors
Addressing patients request to add dietary supplements in cancer care- A suggested approach.
Review 29Nov2023.
Overall, this is an interesting synthesis of the different perspectives of patients and healthcare providers on the complexity of the field of dietary supplements. They also suggest a tool to facilitate patient-centered communication and informed decision making. They provide a comprehensive overview that could be interesting and relevant to daily clinical practice. However, there are a few areas of minor concern that require attention.
1. Abstract: This correctly reflects the content of the full text. The background could be summarized (lines 11-22) and more emphasis should be placed on the tool proposed by the authors (Line 24) for example highlighting reliable sources, risk stratification of the supplement, specific recommendations and subsequent monitoring.
2. Introduction: In general, the introduction is extensive and could be more focused on sections related to the main data, summarizing, and reducing examples while maintaining the same idea and structure: current evidence, patient perspective and new lines of research. It might be more precise to substitute some subjective term and go directly to the objective data that the authors describe correctly. Examples:
- Line 45: “patients tend to shy away” (Is there any data on the % of patients who avoid discussing this topic with their physicians?).
- Line 56: “Regrettably”.
- Line 68: “unequivocally”.
- Line 84: “intriguing possibility”
3. Motivations for Supplement Use: I recommend to group and summarize these motivations in a descriptive way and try to avoid subjective terms when possible (example: line 131 “leaving patients feeling adrift in a sea of uncertainty”. It could be directly said that an active participation of the patient can increase their feeling of control. The paragraph in lines 146 to 147 refers to the credibility of the information sources. Perhaps it could be included in section number 3 regarding the challenges to be faced.
4. Developing and Approach: Numbered as section 5 and it is 4. Supporting points repeated in the first paragraph (lines 235-242) that could be omitted or summarized.
In Figure 1, in the last box, I recommend adding two points (a referral to specific team if needed and share learning) as is explained in the text.
5. Recommending Dietary Supplements as Food: Numbered as section 6 and it is 5.
6. Conclusion: Numbered as section 7 and it is 6. A summary could be included with the specific approach that authors propose including reliable sources, risk stratification and specific recommendations as well as subsequent monitoring. If it has been possible to simplify previous sections in the introduction as suggested, perhaps a discussion section could be added.
Round 2
Reviewer 1 Report
Comments and Suggestions for Authors
Thanks for considering all the comments and addressing them.